The evolution of unique cranial traits in leporid lagomorphs

Wood-Bailey Amber P. 1 apwb@liverpool.ac.uk
http://orcid.org/0000-0001-9782-2358 Cox Philip G. 2 3
http://orcid.org/0000-0001-5117-5335 Sharp Alana C. 1
1 Department of Musculoskeletal and Ageing Science, University of Liverpool , Liverpool, Merseyside , United Kingdom
2 Department of Archaeology and Hull York Medical School, University of York , York , United Kingdom
3 Department of Cell and Developmental Biology, University College London, University of London , London , United Kingdom
Carril Julieta
Electronic publication date: 2022 Nov 29
Publication date: 2022
Volume: 10
Electronic Location ID: e14414
Received 2022 May 16; Accepted 2022 Oct 28
Copyright: © 2022 Wood-Bailey et al.
Copyright year: 2022
Copyright holder: Wood-Bailey et al.
License: This is an open access article distributed under the terms of the Creative Commons Attribution License, which permits unrestricted use, distribution, reproduction and adaptation in any medium and for any purpose provided that it is properly attributed. For attribution, the original author(s), title, publication source (PeerJ) and either DOI or URL of the article must be cited.
License URL: https://creativecommons.org/licenses/by/4.0/

Keywords: Morphology, Phylogenetics, Ancestral states, Rabbits, Skulls, Hares, Fossil mammals, Intracranial joint, Cranial kinesis

Funding: UKRI Natural Environment Research Council via PhD DTP This work was supported by the UKRI Natural Environment Research Council via PhD DTP funding. The funders had no role in study design, data collection and analysis, decision to publish, or preparation of the manuscript.

==============================
Background

The leporid lagomorphs (rabbits and hares) are adapted to running and leaping (some more than others) and consequently have unique anatomical features that distinguish them from ochotonid lagomorphs (pikas) and from their rodent relatives. Two traits that have received some attention are fenestration of the lateral wall of the maxilla and facial tilting. These features are known to correlate with specialised locomotory form in that the faster running species will generally have fenestration that occupies the dorsal and the anteroventral surface of the maxillary corpus and a more acute facial tilt angle. Another feature is an intracranial joint that circumscribes the back of the skull, thought to facilitate skull mobility. This joint separates the anterior portion of the cranium (including the dentition, rostrum and orbit) from the posterior portion of the cranium (which encompasses the occipital and the auditory complex). Aside from the observation that the intracranial joint is absent in pikas (generalist locomotors) and appears more elaborate in genera with cursorial and saltatorial locomotory habits, the evolutionary history, biomechanical function and comparative anatomy of this feature in leporids lacks a comprehensive evaluation.

Methodology

The present work analysed the intracranial joint, facial tilting and lateral fenestration of the wall of the maxilla in the context of leporid evolutionary history using a Bayesian inference of phylogeny (18 genera, 23 species) and ancestral state reconstruction. These methods were used to gather information about the likelihood of the presence of these three traits in ancestral groups.

Results

Our phylogenetic analyses found it likely that the last common ancestor of living leporids had some facial tilting, but that the last common ancestor of all lagomorphs included in the dataset did not. We found that it was likely that the last common ancestor of living leporids had fenestration that occupies the dorsal, but not the anteroventral, surface of the maxillary corpus. We also found it likely that the last common ancestor of living leporids had an intracranial joint, but that the last common ancestor of all living lagomorphs did not. These findings provide a broader context to further studies of evolutionary history and will help inform the formulation and testing of functional hypotheses.

Introduction

The order Lagomorpha is a geographically widespread mammalian group, with a rich taxonomic history (now somewhat reduced) that dates almost to the Cretaceous–Paleogene (K-Pg) extinction event (Lopez-Martinez, 2008). As herbivores, some of which are adapted to a cursorial locomotory form, lagomorphs have a set of anatomical features that distinguish them from their distant rodent relatives, but these differences did not prompt systematists to grant them ordinal status, separate from rodents, until much later than many other mammalian orders (Gidley, 1912). A general understanding that lagomorphs are morphologically conservative with an “evolutionary picture [that is] one of the simplest of any group of mammals” (Wood, 1957), has somewhat exacerbated the lack of research focusing on the group, relative to rodents. However, the extensive use of the European rabbit (Oryctolagus cuniculus) as a model organism in medical research, particularly research that relates to disease and disorders of the musculoskeletal system (Esteves et al., 2018; Li et al., 2015), warrants further understanding of the general gross anatomy and evolutionary history of lagomorphs as a whole.

The literature on functional anatomy in lagomorphs has primarily focused on the limbs in relation to locomotion (De Bastiani et al., 1986; Bleefeld & Bock, 2002; Camp & Borell, 1937; Fostowicz-Frelik, 2007; Gambaryan, 1974; Petajan, Songster & McNeil, 1981; Young et al., 2014; Wible, 2007; Williams et al., 1998), with comparatively less research having been undertaken on the cranium (Bramble, 1989; Kraatz et al., 2015; Stott, Jennings & Harris, 2010; Watson et al., 2014; Watson et al., 2021). Many cranial features appear to correlate with posture and gait (DuBrul, 1950) and there are a number of unique traits that are poorly understood in terms of how they relate functionally to ecological factors such as diet, locomotion and burrowing (Bramble, 1989; Feijó et al., 2020; Gambaryan, 1974; Kraatz et al., 2015; Moss & Feliciano, 1977; White & Keller, 1984). One interesting cranial feature in leporid lagomorphs (rabbits and hares) is an intracranial joint that may facilitate cranial kinesis. The intracranial joint is located between the parietal and occipital bones dorsally, the basioccipital-basisphenoid ventrally and between the squamosal and otic complex at the sides of the cranium (Bramble, 1989). This feature divides the cranium into anterior and posterior units and is thought to provide movement that is distinct from that seen at the other cranial sutures (Bramble, 1989) (Fig. 1G). It is most elaborate in the extant genus Lepus (hares and jackrabbits), although this has not been quantified in terms of complexity or extent of movement. Intracranial joints are common in vertebrates such as reptiles and birds but their presence in leporids is unique for mammals (Bailleul, Witmer & Holliday, 2016; Bock, 1964; Holliday & Witmer, 2008; Iordansky, 2011; Jones et al., 2017). However, there is evidence of a similar structure (a fenestrated mid-cranial gap) in the pachyrukine notoungulates that may have functioned in a similar way (MacPhee, 2014). In other animals, intracranial joints span a wide range of joint types and functions primarily in feeding (e.g., Dutel et al., 2015; Holliday & Witmer, 2008). In leporids, the function is unclear and the comparative evolutionary, histological and biomechanical data needed to fully understand it is lacking (Bramble, 1989). Furthermore, the influence of other ecological factors, such as diet, has not been sufficiently explored.

Figure 1 Morphological characters added to the matrix developed by Meng, Hu & Li (2003), Asher et al. (2005) and Rose et al. (2008).

(A–C) The angle of the upper diastema to the line of the occipital plane is shown here on example Ochotona, Caprolagus and Lepus specimens to illustrate the three states in this character. In Ochotona, there is no facial tilting of note (state 0), in Caprolagus, there is moderate facial tilting (state 1) and in Lepus, there is more extreme facial tilting (state 2). (D–F) The morphological differences in maxillary fenestration are shown in Ochotona, Caprolagus and Lepus. In Ochotona, a single vacuity appears in the posterodorsal part of the maxillary corpus, highlighted in red (state 0), in Caprolagus, a latticework of small openings are restricted to the dorsal part of the maxillary corpus, highlighted in orange (state 1) and, in Lepus, fenestrations are located in the dorsal as well as the anteroventral surface of the maxillary corpus, highlighted in yellow (state 2). (G) The intracranial joint can be seen here between the supraoccipital and parietal bones in the lateral and posterior view, between the basioccipital and basisphenoid in the ventral view. In non-leporid lagomorphs, the joint is not present.

Other unique features of the leporid cranium that have had more recent attention include fenestration of the wall of the lateral maxilla and the presence of marked facial tilting. Maxillary fenestration appears in all leporid genera, albeit to varying degrees (Moss & Feliciano, 1977; Watson et al., 2021). Ochotonids, the sister-group to leporids, also share this trait; however, in Ochotona, it presents as a single vacuity (Moss & Feliciano, 1977). There are two primary hypotheses regarding the function of this trait: the first postulates that it serves to lighten the cranium, reducing torque forces during high-speed locomotion (DuBrul, 1950) and the second, is that it relates to the lack of masticatory forces transmitted through the area (Moss & Feliciano, 1977). However, a recent biomechanical investigation of the strains generated during mastication has shown that fenestrations do not diminish the transmission of masticatory forces, and therefore likely supports the first hypothesis: minimising bone mass while maintaining a mechanically resistant morphology (Watson et al., 2021). Facial tilting in leporids was identified by Kraatz et al. (2015) who noted that there is variation in the angle between the upper diastema and occipital plane across leporids. They hypothesised that this functions to increase frontation of the orbits in order to aid vision in taxa that have specialised, high-speed locomotion. The presence and complexity of these cranial specialisations have been found to vary with locomotory form; for example, the fastest running species have the greatest degree of fenestration in their crania (and subsequently, markedly lighter skulls) (Bramble, 1989; DuBrul, 1950), higher degrees of tilting in the facial region (Kraatz & Sherratt, 2016) and more elaborate intracranial joints (Bramble, 1989). As there appears to be a correlation between increased facial tilt and fenestration of the lateral wall of the maxilla, it is somewhat likely that they form a functional complex that allows the cranium to withstand the mechanical forces present during high-speed locomotion.

Lagomorphs are notable in exhibiting higher diversity in the fossil record than today with 12 extant genera (11 leporid, one ochotonid) and ~94 extant species (61–63 leporid, 30 ochotonid) compared to approximately 78 genera and 234 species from the fossil record (Lopez-Martinez, 2008). Due to this, and the conservative lagomorph body plan, relationships between extant taxa and their recent ancestors are not well resolved by morphological data alone. Although large-scale molecular phylogenetic studies have aided the general systematics, the use of morphological data in character-based phylogenetic methods remains important for time calibration, inferring the phylogenetic position for taxa which are not represented by any tissues from which DNA could be extracted, ancestral state reconstruction and trait evolution rates (Donoghue & Yang, 2016). The recognition of additional morphological variation and formulation of new morphological characters would therefore be welcomed (Ruf, 2014). Furthermore, the use of comparative phylogenetic methods in the field of functional anatomy allows for the study of functional traits (or groups of functional traits) in the context of the evolutionary history of a group (Blanke et al., 2017; McElroy, Hickey & Reilly, 2008).

Due to the difficulties of preserving the anterior and posterior portions of the cranium together in situ, as the two parts separate easily during the taphonomic process, the posterior portion of the cranium is often poorly preserved or entirely absent in many fossil remains (Quintana, Köhler & Moyà-Solà, 2011). This complicates any attempt to confirm the presence or absence of an intracranial joint in extinct species. Furthermore, the identification of facial tilt angle and fenestration of the wall of the lateral maxilla also requires high levels of cranial preservation in the fossil record. However, by applying ancestral state reconstruction to a morphological discrete character matrix, which includes data from extant and extinct species such as Palaeolagus, it is possible to predict the most likely character state at internal nodes on the resultant phylogenetic tree (Reyes et al., 2018).

There are two specific aims of this work. First, to recover a robust phylogenetic topology using morphological characters and generated via Bayesian inference. Secondly, to utilise this tree to undertake an ancestral state reconstruction to better understand where these unique cranial traits mentioned above likely arose in the lagomorph lineage. Results from this work will be used to ascertain whether any of these unique cranial traits would be useful as morphological characters for leporid systematics in general.

Methodology

Phylogenetic matrix

The matrix used was primarily based on that published by Asher et al. (2005), which is based on a matrix developed by Meng, Hu & Li (2003) supplemented with additional characters. Character definitions for original characters are identical to those in Asher et al. (2005). As well as adding characters, the number of taxa was expanded to more comprehensively sample extant species diversity. The resulting morphological data matrix contains 23 taxa and 228 characters. The ingroup taxa are extant and extinct lagomorphs with a tree-shrew genus Tupaia serving as the outgroup.

Additional taxa

The matrix developed by Asher et al. (2005) includes a broad range of both extant and extinct genera belonging to, or closely related to, the supraorder Glires. The taxa used by Asher et al. (2005) were chosen to place new Gomphos material in the context of Glires systematics. For extant lagomorphs, the original inclusion of Lepus, Sylvilagus, Oryctolagus and Ochotona was expanded to include all extant genera and multiple species for genera that are polyspecific (Table 1). For fossil data, only genera that are part of, or closely related to, the lagomorph lineage were included (Mimolagus, Gomphos, Mimotona, Paleolagus, Prolagus).

Table 1 The genera included in previous datasets (Meng, Hu & Li 2003; Asher et al., 2005; Rose et al., 2008) vs the genera and species included in the present study.

The new dataset adds a member of every extant lagomorph genus and multiple species for genera that are polyspecific. Specimen codes relate to the specimen number specific for the Museum collection.

Genera incl. in previous datasets	Genera and species incl. in present study	Specimens used to score characters	Literature used to score characters	Source of specimen	
Lepus	Lepus californicus
Lepus timidus
Lepus europaeus
Lepus americanus	imnh:r:73
amnh:mammals:M-18300
dmet:LE1
amnh:mammals:97648	Ward Lyon (1904)	Idaho MNH via Morphosource
Liverpool World Museum
University of Hull via Morphosource
AMNH Mammology via Morphosource	
Oryctolagus	Oryctolagus cuniculus	l-cet:021			
Sylvilagus	Sylvilagus bachmanii
Sylvilagus audubonii	mvz:mammal:specimens:mvz:mamm:228957
LACM:Mammals:34346	Ward Lyon (1904)	MVZ Arctos via Morphosource
LACM:mammology via Morphosource	
	Brachylagus idahoensis	amnh:mammals:92869		AMNH Mammology via Morphosource	
	Bunolagus monticularis	mcz:mamm:56905	Ward Lyon (1904),
Ge et al. (2015)	MCZ Mammology via Morphosource	
	Caprolagus hispidus	nml:15.5.60.29		Liverpool World Museum	
	Pentalagus furnessi	NSMT:M:42893	Ward Lyon (1904)	National Museum of Nature and Science, Tokyo via Morphosource	
	Poelagus marjorita	LACM:Mammals:14472	Ward Lyon (1904)	LACM:mammology via Morphosource	
	Romerolagus diazi	AMNH:Mammals:M-148181	Ward Lyon (1904)	AMNH Mammology via Morphosource	
	Pronolagus crassicaudatus	nml:A20.11.1908.3		Liverpool World Museum	
Ochotona	Ochotona pallasi
Ochotona princeps	amnh:mammals:59712
amnh:mammals:120698		AMNH Mammology via Morphosource	
†Mimolagus	†Mimolagus	Scored characters pre-existing			
†Gomphos	†Gomphos	Scored characters pre-existing			
†Mimotona	†Mimotona	Scored characters pre-existing			
†Palaeolagus	†Palaeolagus		New characters scored using Wolniewicz & Fostowicz-Frelik (2021)		
†Prolagus	†Prolagus sardus		New characters scored using Dawson (1969)		
Tupaia (outgroup for Glires)	Tupaia (outgroup for Glires)	Scored characters pre-existing			

Additional characters

Three new characters were added to the matrix (Fig. 1). These characters represent cranial traits that are potentially linked to locomotory habit as described above.

Character 95: angle between the upper diastema and the occipital plane (facial tilt) (Kraatz et al., 2015)—(0) more obtuse: >50° (e.g., Ochotona pallasi) (Fig. 1A), (1) moderate tilt: between 40–49° (e.g., Caprolagus hispidus) (Fig. 1B), (2) more acute: <40° (e.g., Lepus capensis) (Fig. 1C).

The non-leporids included in the taxon list are defined as having less facial tilting, i.e., more obtuse facial tilt angles (more than 50°). For leporids, all of whom have some facial tilting, a species is defined by having a moderate facial tilt if the median angle between the upper diastema and the occipital plane is between 40–49°, and a more extreme facial tilt if the median angle between the upper diastema and occipital plane is less than 40°. This character is based primarily on data collected by Kraatz et al. (2015) who found that leporid species exhibiting a specialised mode of locomotion, cursorial or saltatorial, generally exhibit a more acute degree of facial tilting than those who exhibit generalist locomotion.

Character 113 lateral fenestration of maxilla (if present)—(0) large single opening occurring in the posterodorsal part of the maxillary corpus (e.g., Ochotona pallasi) (Fig. 1D), (1) a latticework of small openings restricted to the dorsal part of the maxillary corpus (reduced) (e.g., Caprolagus hispidus) (Fig. 1E), (2) an extensive latticework of small openings occupying the dorsal as well as the anteroventral surface of the maxillary corpus (advanced) (e.g., Lepus timidus) (Fig. 1F).

A multi-state character was necessary to expand on the original character for lateral fenestration of the wall of the maxilla (character M121, MW66, A111 in Supplemental Document 2) as the differences appear to correlate with locomotory form. This character is coded as inapplicable for those without lateral fenestration of the wall of the maxilla.

Character 136 (new) intracranial joint—(0) absent (e.g., Ochotona pallasi), (1) present (e.g., Lepus capensis) (Fig. 1G).

The intracranial joint is located along the occipito-parietal union dorsally and continues down either side of the braincase between the squamosal and the otic complex. A number of rabbit genera (including Oryctolagus, Brachylagus and Sylvilagus) feature an unfused interparietal bone and so the joint is diverted around the posterior edge of the interparietal. Mid-ventrally, it is completed by a union at the basioccipital-basisphenoid articulation (Bramble, 1989). Due to the lack of data pertaining to the variation in complexity of this feature between genera and species, there is no justification for a multi-state character. Therefore, it is coded as absent or present.

Character 173 (new) interparietal state (if present)—(0) unfused (e.g., Oryctolagus), (1) fused (e.g., Lepus).

A fourth, new character was added to the dataset. This character expands upon “interparietal occurrence” to account for diversity in interparietal fusion. This successfully splits Lepus from some rabbit genera (Sylvilagus, Oryctolagus, Romerolagus, Brachylagus and Nesolagus). However, this trait is not looked at further using ancestral state reconstruction.

Four soft-tissue characters were removed from the dataset of Asher et al. (2005): trophoblast (A225), maternal-fetal nutrient exchange (A226), gestation time relative to body mass (A227) and ureter (A228).

Phylogenetic approach

A relaxed clock analysis was implemented using a fossilised birth-death model in the program Mr Bayes v. 3.2.7a (Ronquist et al., 2012) via CIPRES Science Gateway (Miller, Pfeiffer & Schwartz, 2010). Some groups were constrained (using prset tologypr=constraints) in order to better fit the topology of published trees which used molecular data (Ge et al., 2013; Matthee et al., 2004). In this instance, constraining was justified as the difficulty of producing accurate topologies from morphological data for lagomorphs is well reported (Kraatz et al., 2021). The fossil taxa were calibrated by age of fossil occurrence (via fossilworks.org) and a soft upper bound constraint was placed on the age of the tree (prset treeagepr=offsetexp) based on the molecular estimate of the age of Mimotona (the stratigraphically oldest taxon included in the analysis) (dos Reis, Donoghue & Yang, 2014). A calibration was also placed on the age of the genera Lepus, Ochotona and Sylvilagus based on the posterior distribution of the divergence estimates from Matthee et al. (2004) (Table 2). The strategy under which the species were sampled was set to represent all major lineages (diversity sampling) (prset samplestrat=diversity). The base of the clock rate was set using an informative prior derived from a non-clock analysis of the dataset (prset clockratepr=lognorm). The clock model for rate variation among lineages was set to a relaxed uncorrelated clock with values sampled from a gamma distribution (IGR). Six MCMC chains were run twice for 7,000,000 generations and sampled every 1,000 generations. The first 25% of each run were discarded at the burnin phase.

Table 2 The constraints and calibrations placed on clades so that the reconstructed topology and divergence time estimates of our tree is more concordant with published molecular phylogenies.

Constraint	Taxa	Divergence calibration	
Ingroup	All taxa bar Tupaia	N/A	
Lepus	L. californicus, L. timidus, L. europaeus, L. americanus	4.03–5.90 (Matthee et al., 2004)	
Ochotona	O. pallasi, O. princeps	23.31–39.26 (Matthee et al., 2004)	
Sylvilagus	S. bachmanii, S. audubonii	2.43–6.65 (Matthee et al., 2004)	
Leporids	All Lepus sp., B. idahoensis, B. monticularis, C. hispidus, N. timminsi, P. furnessi, P. marjorita, R. diazi, P. crassicaudatus, O. cuniculus, S. audubonii, S. bachmanii	N/A	
Clade_one	O. pallasi, O. princeps, P. sardus	N/A	
Clade_two	N. timminsi, P. marjorita, P. crassicaudatus	N/A	
Clade_three	C. hispidus, O. cuniculus, B. monticularis, P. furnessi	N/A	
Clade_four	R. diazi, B. idahoensis, S. audubonii, S. bachmanii	N/A	

Reconstructing ancestral states

Due to incomplete preservation of the cranium it is often difficult to ascertain the state of the new cranial characters in fossils, particularly in very old specimens. It can be difficult to reliably measure facial tilt angle in fossils because specimens are likely to be deformed due to over-laying rock. Similarly, the struts that are characteristic of fenestration of the lateral maxilla can be damaged easily (although the general area of fenestration remains). Finally, it is not justified to make any claims to a kinetic intracranial joint in fossil leporids based on anatomy alone, as these may represent intermediate forms. Ancestral state reconstruction allows for the combination of observed character state data at the tips of a tree and information regarding the phylogenetic relationships between taxa—resulting in the ability to predict states of characters at internal nodes (Holland et al., 2020). Ancestral state reconstructions were undertaken in the R-language toolkit MBASR (MrBayes Ancestral States with R) (Heritage, 2021; R Studio Team, 2020). This toolkit performs ancestral state reconstruction using the continuous-time Markov model via MrBayes and automates many of the steps included in packages with similar functions (Heritage, 2021).

The consensus tree from the relaxed clock analysis was loaded into MBASR with a file including the specific trait data examined. The number of samples generated was set at 10,000 following a sensitivity analysis with lower values. MBASR applies a likelihood filter (the threshold for this filter is 25% of the likelihood range) and so this value allows enough generations to reach optimum proposals in terms of likelihoods. Each run reconstructed the ancestral states for a single character.

The characters for which ancestral state reconstruction was performed were: ch. 95 (facial tilt), ch. 113 (fenestration) and ch. 136 (intracranial joint). Character 113, relating to the degree of fenestration of the lateral wall of the maxilla, is eligible for ordering (as there is good evidence for a progression of states). This was tested and ordering the states was found to make very little difference to results.

Results and Discussion

Phylogeny

The relaxed-clock phylogenetic reconstruction was derived from morphological data and includes members of every extant genus of lagomorph (Supplemental Document 4). Previous studies of lagomorph phylogeny have yielded contrasting topologies, our analysis is no exception and differs in several respects (Fig. 2). Our results agree with Matthee et al. (2004) and Ge et al. (2013) in finding Nesolagus and Pronolagus to be closely related, though we do not consider them sister taxa (as in Ge et al. (2013)), instead finding Nesolagus basal to a clade comprising Pronolagus and Poelagus. Furthermore, both prior analyses found this clade to sit at the base of Leporidae, while we recover it as more highly nested, and consider Lepus to be the most basal leporid. We find derived leporids are split into two subclades, one comprising Bunolagus, Oryctolagus, Caprolagus and Pentalagus and a second containing Brachylagus, Sylvilagus and Romerolagus. Matthee et al. (2004) found that our second clade formed a paraphyletic assemblage, which they also considered monophyletic, albeit with different interrelationships. Direct comparison with Ge et al. (2013) is complicated by the unresolved polytomy located towards the base of this group. However, any monophyletic resolution of the first clade would necessarily include Sylvilagus and Romerolagus, and therefore differ from our own topology.

Figure 2 The phylogenetic hypotheses of extant Lagomorpha.

(A) Adapted from Matthee et al. (2004), (B) Ge et al. (2015) and (C) our phylogeny.

The divergence time estimates in our phylogenetic reconstruction roughly match those in molecular studies, with a key difference being the divergence estimate in the leporid/ochotonid split. For this, Ge et al. (2013) gave a median value of divergence time as 50.3 million years and Matthee et al. (2004) gave 31.7 million years. Our phylogeny gives a median estimate of 27.3 million years. The estimates for the divergence of leporids are well-constrained in comparison to the leporid/ochotonid split age estimations (20.2 Mya here, 15.2 Mya for Matthee et al. (2004) and 18.1 Mya for Ge et al. (2013)).

Clade groupings for extant lagomorphs are notoriously difficult to resolve (using molecular or morphological data) due to morphological conservatism, the absence of chromosomal synapomorphies and the saturation of mitochondrial DNA sequences (Matthee et al., 2004). Given that we used morphological data alone, it was necessary to provide the model with information derived from molecular phylogenies. The relevant divergence time estimates in our phylogenetic reconstruction generally fall within the published ranges, with the exception of the leporid/ochotonid split in Ge et al. (2013). This was largely aided by the calibration of the age of the genera Lepus, Ochotona and Sylvilagus. Without these calibrations, the divergence time estimates are far younger than expected; for example, a phylogenetic reconstruction with no additional calibration of certain genera places the divergence between leporids and ochotonids at around 10.9 million years. This reflects the young estimates for clade divergence that morphological data alone, with a poor sampling of fossil specimens, tends to produce (Barba-Montoya, Tao & Kumar, 2021). By placing a few key calibrations on large extant genera, we compute a tree with estimations that are concordant with previous studies.

Ancestral state reconstruction

The results from the first reconstructed trait, the angle between the upper diastema and occipital plane (facial tilting) indicate that facial tilting may have been present in the last common ancestor of the living leporids, although to what extent is unresolved (Fig. 3). The outcome of the ancestral state reconstruction is bolstered by the finding that an early leporid genus, Hypolagus, exhibits a notable degree of facial tilting potentially exceeding any living leporid (Hibbard, 1969). The ancestral state reconstruction for this trait suggests that the last common ancestor of extant lagomorphs, both including and excluding Palaeolagus, likely had no facial tilting beyond that of the other closely related mammals such as rodents. The inclusion of more fossil lagomorphs is needed to determine whether this is a result driven by phylogeny or by locomotory similarities between included fossil groups in our study (such as Prolagus and Palaeolagus). Previous work on leporid facial tilting found that it was strongly homoplastic across leporid evolutionary history and that there was weak phylogenetic signal in the facial tilt angle (Kraatz & Sherratt, 2016; Kraatz et al., 2015). Furthermore, disaggregating the raw data for these angles reveals a large amount of intraspecific variation, in some species, up to 20.2° as in Pronolagus crassicaudatus (Kraatz et al., 2015), suggesting that it is likely a trait driven more by environmental than evolutionary factors. Specifically, Kraatz et al. (2015) found that in generalist locomotors, such as Romerolagus diazi, there is reduced facial tilt angle in comparison to cursorial and, to a lesser extent saltatorial locomotors such as Lepus californicus and Sylvilagus audubonii. This suggests that perhaps locomotion might be a driver for facial tilt angle, rather than phylogeny. Due to the lack of significant phylogenetic signal, high homoplasy and the influence of ecological factors (primarily locomotion), the use of this trait as a morphological character is not recommended. More work is needed to determine key drivers for this trait, including study of this trait in fossil groups.

Figure 3 Ancestral state reconstruction of facial tilt.

All species within the range of the orange line are within Lagomorpha, within the blue line are within Leporidae and within the green line are within Ochotonidae. Red in the nodal markers refers to an obtuse facial tilt (as in extant Ochotona), yellow refers to a moderate facial tilt (as in Caprolagus) and orange refers to an extreme facial tilt (as in most Lepus).

The second reconstructed trait, fenestration of the rostrum, indicates that the intermediate fenestration seen in rabbits such as Oryctolagus, Romerolagus and Nesolagus is the likely ancestral state of leporids and all lagomorphs (Fig. 4). Therefore, the advanced and singular opening states in most Lepus and all Ochotona respectively are likely derived traits. Whilst fossil lagomorph taxa are often only represented by teeth or mandibular sections, well preserved members of Palaeolagus (33.9–20.43 Mya) and Alilepus (13.6–1.8 Mya) appear to also feature the intermediate, rabbit-like, state (Wolniewicz & Fostowicz-Frelik, 2021; Wu & Flynn, 2017), supporting our results. Our ancestral state reconstruction also suggests that the development of the advanced form of fenestration seen in most Lepus and some other taxa, such as Sylvilagus and Brachylagus, has evolved on two separate occasions in the lineages of extant leporids, whereas the single vacuity state seen in ochotonids likely evolved once in the common ancestor of Ochotona and Prolagus. However, in phylogenies reconstructed by Matthee et al. (2004) the genus Lepus is in a clade with Sylvilagus, Brachylagus and other taxa. If we accept their reconstructions as correct, then the advanced fenestration in these taxa would have likely evolved once, in the common ancestor of Lepus, Brachylagus and Sylvilagus.

Figure 4 Ancestral state reconstruction of fenestration of the lateral wall of the maxilla.

All species within the range of the orange line are within Lagomorpha, within the blue line are within Leporidae and within the green line are within Ochotonidae. Red in the nodal markers refers to a single vacuity (as in extant Ochotona), yellow refers to fenestration above the line of the bony remnant of the lacrimal duct (as in Lepus americanus) and orange refers to fenestration above and below the line of the bony remnant of the lacrimal duct (as in Lepus californicus).

Fenestration of the lateral wall of the maxilla is considered a diagnostic feature of the leporid cranium and its state varies between taxa. There are multiple hypotheses as to the function of these fenestrations, including lightening the rostrum for faster running or lack of masticatory force transmission (DuBrul, 1950; Moss & Feliciano, 1977; Watson et al., 2021). Recently, a study utilitising both multibody dynamics analysis and finite element analysis suggested that the fenestration is optimised to reduce mass in the rostrum whilst maintaining structural integrity during mastication (Watson et al., 2021). In this scenario, both primary functional hypotheses (lightening the skull for locomotion and masticatory force response) could be correct. The ancestral state reconstruction presented here does not bolster any functional hypothesis; however, the presence of the advanced fenestrations in taxa that run at slower speeds, such as Brachylagus, which probably locomotes at around 23 km per hour, as opposed to Lepus europaeus (which reaches speeds of 75 km per hour according to Schai-Braun et al. (2015)) suggests that the function is not entirely related to running speed. This trait also needs more study in order to identify the amount of intraspecific variation and measure the extent and complexity of the maxillary fenestrations. This trait could be utilised as a morphological character in further phylogenetic analyses; whilst the original character set included a character for the presence of fenestration in the maxilla, information regarding the degree of the fenestration could help to separate extant taxa further.

The third trait reconstructed at internal nodes, the leporid intracranial joint, is shown mapped on the consensus tree of the relaxed clock analysis (Fig. 5). This suggests that the last common ancestor of all extant leporids likely possessed the joint. This outcome was expected, as we predicted that this trait arose as leporids became more specialised in morphology, possibly relating to the transition from more generalised to more specialised high-speed locomotion (Gambaryan, 1974). The osteological remains of pachyrukine notoungulates feature a similar fenestrated mid-cranial gap and superficially resemble leporid lagomorph morphology in body and cranial shape (MacPhee, 2014). These animals were likely saltators and they filled a similar ecological niche to leporids, indicating that their mid-cranial gap is perhaps a similar adaptation to a similar locomotor mode (Seckel & Janis, 2008). The ancestral state reconstruction also suggests that it is likely, although with less certainty, that the last common ancestor of all extant lagomorphs did not have this trait. This outcome was also expected given a close relative of this ancestor, Palaeolagus, exhibited a generalist locomotory form and appears to have a posterior cranium that resembles ochotonid morphology (Wolniewicz & Fostowicz-Frelik, 2021). For large-scale phylogenies, where distinguishing between leporids and ochotonids is necessary, the presence/absence of an intracranial joint could be a useful morphological character. However, it is rarely preserved in fossil taxa and in the character’s current state (presence/absence), it does not provide any means of differentiating between extant leporid taxa. Future work on the variation of this trait among leporids may allow us to categorise different degrees of complexity, aiding our ability to use this feature as a useful character in leporid systematics, and potentially identifying links to locomotion.

Figure 5 Ancestral state reconstruction of the presence of an intracranial joint.

All species within the range of the orange line are within Lagomorpha, within the blue line are within Leporidae and within the green line are within Ochotonidae. Red in the nodal markers refers to the absence of an intracranial joint (as in all ochotonids). Orange refers to the presence of an intracranial joint (as in all leporids).

Conclusion

This study found that the last common ancestor of extant leporids likely had an intracranial joint, but that the last common ancestor of extant lagomorphs likely did not—indicating that this trait was potentially driven by changes in locomotory form in the early leporids (from generalist to more specialist). It also found that the ancestral state of maxillary fenestration was likely the intermediate rabbit-like form, with the extreme advanced and singular forms in Lepus and Ochotona representing derived features. The ancestral state reconstruction for facial tilting suggests that the last common ancestor of living leporids likely had some form of facial tilt, although the extent of which is unresolved. This study also found that the last common ancestor of all lagomorphs likely did not have any facial tilt of note. In future work, broader sampling of fossils is necessary to avoid the need to calibrate clade divergence times, particularly those fossils that are closer to the leporid/ochotonid split. Furthermore, the study of these traits would benefit from a total evidence approach, combining molecular and morphological characters, to ensure the accuracy of resolved phylogenetic relationships.

Supplemental Information

Supplemental Information 1 The phylogenetic matrix used in the study.

Click here for additional data file.

Supplemental Information 2 Descriptions of characters used in the phylogenetic matrix.

Click here for additional data file.

Supplemental Information 3 Trait dataset used for ancestral state reconstruction.

Remove titles before attempting to run file

Click here for additional data file.

Supplemental Information 4 Our relaxed-clock phylogenetic reconstruction derived from morphological data.

The tree includes at least one member of every extant genus of lagomorph. Fossil taxa are denoted with “†”. The values at internal nodes are the median divergence time estimates at key nodes. Tupaia and Ochotona silhouettes used under Public Domain Dedication 1.0 license. Sylvilagus and Lepus silhouettes used under Creative Commons Attribution-NonCommercial 3.0 Unported license with credit given to Gabriela Palomo-Munoz and Sarah Werning.

Image credits: Phylopic: T. Michael Keesey (after Joseph Wolf), Public Domain Dedication 1.0 (https://creativecommons.org/publicdomain/zero/1.0/): http://phylopic.org/image/88a07585-846a-405d-9195-c15c010e7443/; Margot Michaud, Public Domain Dedication 1.0 (https://creativecommons.org/publicdomain/zero/1.0/): http://phylopic.org/name/79021e04-3b8d-4a49-9902-24d744d4af51; Gabriela Palomo-Munoz, Creative Commons Attribution-NonCommercial 3.0 Unported (https://creativecommons.org/licenses/by-nc/3.0/): http://phylopic.org/image/24535087-a458-4d5e-ad59-eba10a4cd080/; Sarah Werning, Creative Commons Attribution 3.0 Unported (https://creativecommons.org/licenses/by/3.0/), http://phylopic.org/image/dea688b6-9168-4e79-a106-366888148eb1/.

Click here for additional data file.

Thanks to Dr Omar Rafael Regalado Fernandez, Samuel Cross and Dr Tiago Rodrigues Simões for graciously answering the author’s queries regarding the phylogenetic analyses swiftly and clearly. We would also like to thank National Museums Liverpool World Museum vertebrate zoology curators Dr John James Wilson and Tony Parker for access to specimens. Thanks also to Dr Brian Kraatz for the discussion at the World Lagomorph Society conference 2022, which ultimately made this manuscript more scientifically rigorous. Thank you to the editor(s) and reviewers for considering our manuscript and providing constructive comments, in particular regarding character definitions.

Additional Information and Declarations

Competing Interests

Author Contributions

Data Availability

The authors declare that they have no competing interests

Amber P. Wood-Bailey conceived and designed the experiments, performed the experiments, analyzed the data, prepared figures and/or tables, authored or reviewed drafts of the article, and approved the final draft.

Philip G. Cox conceived and designed the experiments, authored or reviewed drafts of the article, and approved the final draft.

Alana C. Sharp conceived and designed the experiments, authored or reviewed drafts of the article, and approved the final draft.

The following information was supplied regarding data availability:

The raw data is available in the Supplemental Files.

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
