# Peer review of "The evolution of unique cranial traits in leporid lagomorphs"

_PeerJ, doi:10.7717/peerj.14414_

## Round 0.1 · original submission · Minor Revisions

I received two mostly positive reviews of your manuscript. However, it still needs considerable work in order to make the manuscript suitable for publication.

Both reviewers make a detailed revision of your manuscript. Please, take these suggestions into account to improve the quality of your work.

Reviewer 1 ·

Basic reporting

Wood-Bailey and colleagues perform ancestral state reconstruction on a phylogenetic hypothesis for Lagomorpha in order to reconstruct the morphology of three cranial features characteristic for leporid lagomorphs – facial tilt, maxillary fenestration and the intracranial joint – at key nodes in the phylogeny. In general, the study is well executed – the introduction provides sufficient background and context, the applied methodology is appropriate, the description of the study protocol is clear and the results are presented and discussed in an informative way. The study does not contain any major flaws, but the manuscript suffers from several shortcomings that affect its quality. If the authors are able to revise the text according to the presented suggestions which constitute minor revisions, I can recommend publication of their manuscript in PeerJ.

Experimental design

With respect to methodology, one of the characters introduced by the authors – character 113 – is problematic. The authors provide the following definition for character 113: ‘lateral fenestration of maxilla (if present)--- (0) large single opening (e.g., Ochotona), (1) a latticework of small openings (reduced) (e.g., Oryctolagus), (2) a latticework of small openings (advanced) (e.g., Lepus)’. The authors also provide a figure (Fig. 1B) to illustrate the introduced character states. However, the distinction between states 2 and 3 is not clear – defining the states as ‘reduced’ and ‘advanced’ is insufficient to allow future authors to easily understand the differences between these states and the differences between the states are also not clearly visible in the provided figure. Furthermore, because the latticework in the lateral wall of the maxilla in leporids is very delicate, its morphology can also be affected by preservation not only in fossil taxa, but it can also be easily damaged in modern zoological specimens. The authors are asked to consider whether the character states they constructed can be affected by preservation and consider using only two states to describe it – (0) large, single opening and (1) latticework of small openings (following Wible [2007:char. 8]). If the authors decide to reduce the number of states for this character, they should update their ancestral state reconstruction analysis accordingly. However, if the authors insist on retaining three states for this character, the derived character states should be described in more detail and illustrated using close-up photos of actual zoological specimens, instead of line drawings.

Another issue with the methodology is the lack of information on the literature and fossil/zoological specimens used to score the characters for the phylogenetic matrix. The authors are requested to insert an extra column or two into Table 1 and provide this information. It is also unclear why the authors use ‘zz’ to denote extinct taxa in their figures and tables, instead of the more widely used ‘†’. It is requested the authors amend the symbol they use to denote extinct taxa to improve the clarity of their manuscript.

Validity of the findings

no comment

Additional comments

Even though the manuscript focuses on ancestral state reconstruction of three cranial features of leporid lagomorphs, the abstract focuses only on providing context and reporting analytical results regarding one of these features (intracranial joint) and mentions the two other features (facial tilting and maxillary fenestration) only in passing. The authors are requested to re-write the abstract so that it provides context and summarises ancestral state reconstruction results for all three anatomical features.

In comparison to their sister group (Rodentia), and several other clades of mammals and tetrapods more generally, lagomorphs are actually not that ‘successful’ (lines 44–46) – lagomorphs have less than 4% the number of species as rodents and are much more restricted than rodents in terms of morphology, locomotion and habitat range (e.g. Feldhamer et al. 2020). The authors are requested to change the opening sentence of the introduction and present a more realistic summary of the order.

The language and style used throughout the manuscript is not uniform and often falls below a satisfactory standard – some of the anatomical and biological terms used in the text are inappropriate and several statements made by the authors are not specific enough, which hinders the clarity of the writing. There are also some issues with using punctuation marks and formatting, such as errors regarding the correct use of hyphens and en-dashes. The in-text citations do not follow a consistent format and there are problems with the use of brackets. The reference list at the end of the manuscript does not follow a consistent theme and several citations contain errors. I have highlighted numerous errors regarding the use of technical vocabulary, as well as some other issues, and request the authors to amend the manuscript accordingly, but formatting mistakes are not highlighted and the authors are asked to carefully read the manuscript before re-submission in order to improve the quality of the text.

Regarding the figures, the position of the intracranial joint should be clearly marked with a different colour or with an arrow in Fig. 1 – the caption describes the position of the joint relative to the cranial bones, but since the bones are neither labelled nor even visible in the line drawing, the caption will not be informative for potential readers. Furthermore, the entire figure, not just the ‘angle of the upper diastema to line of the occipital plane’ is ‘illustrated in black’ – please be more precise and amend the caption accordingly. Fig. 2 can be completely discarded from the manuscript as it represents the raw output from the phylogenetic analysis, and appropriately edited versions of the cladogram are also presented in Figs 3–6. The expanded captions for Figs 4–6 are of an unacceptable standard as they contain inappropriate anatomical terminology (description of a fenestra/opening as a ‘paucity’; Fig. 5) and describe the content of the figures in an unintuitive way (‘The orange bracket encompasses all lagomorphs, the blue surrounds leporid lagomorphs and in green, the ochotonids’). The authors are requested to shorten the captions for Figs 4–6 and or/re-write them so that they are more informative. Finally, Fig. 1 should be divided into five (A–E) instead of two (A–B) panels, and Fig. 3 should be separated into three (A–C) panels for clarity (with references to these figures throughout the text updated accordingly).

List of minor issues:

Lines 56–59. The authors state that ‘comparatively little research has been undertaken on the cranium’ with regards to locomotion in lagomorphs, in contrast to the amount of research done on the postcranial skeleton. However, the authors actually provide more references for work done on the cranium (5 studies) than on the postcranial skeleton (4 studies)!

Line 67. Please provide a reference to a study clearly illustrating the anatomy of the intracranial joint and its position relative to the cranial bones (e.g. an anatomical atlas, textbook, etc.) which readers can refer to in order understand the morphology of this structure clearly.

Line 71. Please provide a few references to support the claim that ‘intracranial joints are common in (…) reptiles and birds’.

Lines 70–73. The authors state that ‘in other animals, intracranial joints span a wide range of joint types and functions primarily in feeding’ but provide only one reference to a study on dinosaurs. Including more references at least to some other tetrapod or vertebrate groups would provide more context.

Line 75. Please replace ‘have’ with ‘has’.

Line 78. Please be more specific – instead of writing about the ‘fenestration of the lateral maxilla’, write ‘fenestration of the lateral WALL OF THE maxilla’.

Lines 78–81. The morphology of the maxillary fenestration in leporids is summarised as representing ‘varying degrees’, without describing its exact morphology. Please add more information on the morphology of the maxillary fenestration in leporids in order to contrast it with the state present in ochotonids (‘single vacuity’).

Line 80. Please use ‘sister-group’ instead of ‘sister-family’.

Line 82. Replace ‘crania’ with ‘cranium’.

Line 85 – Replace ‘shows’ with ‘has shown’.

Line 87 – Be more specific, as it is not clear what ‘minimising bone’ means. Minimising bone mass? Surface area? Volume?

Lines 98–99. Please replace the term ‘extant biosphere’ with a more appropriate phrase (e.g. ‘today’).

Lines 102–105. Most importantly, morphology-based phylogenetics are crucial for inferring the phylogenetic position of stem-lagomorphs, not represented by any soft-tissue material suitable for DNA extraction.

Lines 105–107. The authors use the phrase ‘identification of new, derived morphological characters’ – but do they actually mean recognition of additional morphological variation and formulation of new morphological characters? Only a character state, not a character itself, can be plesiomorphic or derived, but this should be determined via phylogenetic analysis, not a priori assumptions.

Line 111. ‘Due to the difficulties OF preserving…’

Line 112. ‘separate easily POST-MORTEM/BEFORE BURIAL/etc.’; furthermore, please replace ‘crania’ with ‘cranium’.

Line 114. Please replace ‘directly classify’ with ‘confirm the presence or absence of’.

Line 116. ‘HOWEVER, by applying…’. In addition, the authors mention applying ‘various methods’ – but do not list them, so do they actually mean applying ancestral character state reconstruction only?

Line 121. Generating the phylogenetic hypothesis using Bayesian methods is a means to achieving the aim of providing a robust phylogenetic topology, not the aim itself.

Line 130. Please change the heading to ‘Phylogenetic matrix’.

Lines 131–132. Remove the first sentence as it is more appropriately repeated elsewhere (lines 183–184).

Line 135. ‘were identical to THOSE IN Asher et al. (2005)’

Line 136. Please replace ‘better reflect’ with ‘more comprehensively sample’.

Line 137. Please replace ‘resulting’ with ‘ingroup’.

Lines 142 and 147. ‘closeLY RELATED to’

Lines 152–153. Please replace ‘perform somewhat together’ with ‘are likely functionally linked’.

Line 155. Using a threshold of 39.9 degrees seems very meticulous… Could the authors set the threshold to 40 degrees for convenience?

Lines 162–164. I have never encountered describing animal locomotion in the way the authors describe it in the submitted manuscript, e.g. ‘species that locomote in a saltatorial way’ – perhaps the authors could revise the phrasing and use more widely used phrases such as ‘saltatory species’, ‘species exhibiting a saltatory mode of locomotion’, etc.

Lines 170–171. The abbreviations M121, MW66 and A111 are not explained anywhere in the text.

Lines 180. I do not understand what the authors mean when they say that the character ‘states are currently only observational’ – are not all discrete character states assessed via observation?

Line 191. ‘…the STRATIGRAPHICALLY oldest taxon INCLUDED IN THE ANALYSIS…’

Line 193. ‘…Mathee ET AL. (2004)…’

Line 202. ‘Reconstructing ancestral stateS’

Lines 203–204. The sentence is overly complicated. ‘Due to incomplete preservation of the posterior cranium in fossils, it is difficult to determine the presence or absence of the intracranial joint in extinct taxa’ will suffice.

Line 205. ‘…observerd CHARACTER state data…’

Line 207. Just use ‘characters’ instead of ‘heritable traits’.

Line 216. ‘…reconstructed THE ANCESTRAL STATES FOR a single character state…’

Line 217. ‘The characters for which ancestral state reconstruction was performed were as follows…’

Lines 225–233. The description of differences between the cladograms is not exactly clear and lacks vocabulary usually used in discussion of cladogram topologies. The authors are asked to revise this paragraph to make it more clear and the terminology used more appropriate.

Line 239. Replace ‘here’ with ‘in this study’ and ‘for’ with ‘in’.

Line 244. ‘Give the model’ is not very technical – consider replacing with ‘provide the model with’.

Line 249. ‘it places’ – what is ‘it’?

Lines 257–58. ‘MBASR could not confidently assert one way or another to the state’ – do you mean that it was not possible to reconstruct the character state with some likelihood?

Line 265. I have never come across referring to any group of tetrapods as ‘generalised locomotors’ – please revise this (see also comments for lines 162–164).

Line 283. Replace ‘between’ with ‘of’.

Lines 297–299. Provide references for the cited speed estimates/measurements.

Line 298. Again, please revise ‘locomotes’.

Line 313. Again, ‘generalised locomotor’.

·

Basic reporting

The writing level is appropriate, the literature review (with one exception) is adequate, PeerJ policies on data have been observed. A few usage suggestions are made on the attached PDF. My comments are limited to morphology.

The paper concerns several highly apomorphic features of the leporid cranium that have been extensively discussed in the primary morphological and paleontological literature. The focus of the paper is an effort to establish when and in what context these features might have appeared in lagomorph phylogeny, and whether there are any correlations among them that might point to function. The authors note that “the evolutionary history, biomechanical function and comparative anatomy of [these] feature[s] in leporids lacks a comprehensive evaluation”. Strictly speaking, the evaluation this paper offers is a series of character analyses and time trees based on their observations on extant lagomorphs, with some commentary on the ochotonid and leporid fossil record based on the work of others. The results are in line with the existing literature in regard to the intracranial joint standing as a derived feature of leporids not shared with ochotonids. The authors’ contribution is to pin a Bayesian probability estimate on the possible time of appearance. The absence of a strong correlation between the intracranial joint and head tilt or rostral fenestration is interesting and again in line with recent fossil-oriented observations.

Although the evaluation of their proposed character states in is very complete with regard to extant lagomorphs, it appears that no fossils have been examined by the authors themselves. This seems a pity, because some experience with well preserved remains like those of Palaeolagus may provide morphological insights that paleontologists might have missed. It may also help make the definition of their candidate characters/states more comprehensive.

This statement is incorrect: “Intracranial joints are common in vertebrates such as reptiles and birds but its presence in leporids is unique for mammals.” The skulls of extinct pachyrukine notoungulates exhibit a midcranial fenestration that resembles the intracranial joint of leporids. MacPhee (2014) described associated osteological features in some detail but did not attempt to correlate the fenestration with probable locomotor behavior. The authors are in a good position to comment on this, and to do so would broaden the appeal of their paper.


The paper would benefit from better illustrations of the morphological features of interest, especially with regard to alternative character states. This would also make the paper more useful to readers who might want to employ the authors’ characters in their own work. Figure 1 is especially in need of attention: the intracranial joint is not even identified as such.

MacPhee RDE (2014) The serrialis bone, interparietal complex, “x” elements, entotympanics, and the composition of the notoungulate caudal cranium. Bulletin of the American Museum of Natural History 384: 1-69.

Experimental design

No comment

Validity of the findings

Suggestions made above

Additional comments

none

·

Basic reporting

The three new characters are listed in the following order in the text: (95) Facial Tilt, (113) Fenestration, and (136) ICJ, but that does not seem to be the order they are listed in the supplemental data. I think those, in supplemental order, are listed as ICJ, FT, and then FT. It was difficult to sort how species were coded for these characters. Apologies if I am mistaken.
Given the focus of these manuscript on three key characters that have not been widely considered in such studies, I think it is important that the characters are better described and illustrated. Facial tilt seems sufficiently described and illustrated, but both the ICJ and Fenestration need better description and illustration. See my additional comments below.

Experimental design

The supplemental material (.txt) indicates that four characters from Asher et al. were not used in this study. This should be discussed in the methods section.
I think understanding ancestral states is important, and per my other comments, consider at least discussing Hypolagus, if not incorporating into your phylogenetic analyses.
Facial Tilt Angle
You can probably measure angles from figures for those taxa you did not code (e.g., all fossils) which I would recommend. My other concern is that the FT degree categories do not seem to distinguish among leporids, just between leporids and ochotonids. I realize that this is a continuous character and making it a multistate character might be difficult but consider what angles might distinguish a cursor from a saltator. Also, you should report the angles used here, whether from other sources or those you measured.
Supplemental: Kraatz et al. (2015) reported a FT angle for Nesolagus, but it it was coded as ‘?” in this study. [I think, but I might be confused]

Fenestration
Pentalagus does have fenestration, but I believe is coded as absent. See Ge et al. (2015, http://dx.doi.orgw/10.1163/18759866-08404001)
These are mostly my own observations, so take them as you will. The lacrimal duct, which has a bony remnant, passes dorsoposteriorly to ventrolaterally along the lateral rostrum. This seems to be a major boundary for fenestration patterns. Vacuities (i.e. Ochotona) and reduced fenestration (e.g. Nesolagus) seem to occur only above that bony duct. Whereas significant fenestration occurs below in some taxa (e.g. Oryctolagus). I know that your character is three states, but I think that if you considered it more, you could have a more specific anatomical description than a simple to complex gradient.
Intracranial Joint
I’m glad this character is being considered here. My recommendation is that it is described and illustrated in better detail. Preferably, with photograph images of skulls. I have many photos, just ask if you would like some.
In summary, for Figure 1:
• Make a specific illustration for the ICJ with individual bones identified, consider using a ventral view too.
• For fenestration, identify more specific areas where the fenestration is occurring; vacuity and more limited fenestration occurs more dorsally on rostrum.
• For both ICJ and fenestration, you should use images of real skulls so that those characters can be better observed.

Validity of the findings

Overall, I think the findings are valid, though I note phylogenetic methods are out of the area of my expertise.

Additional comments

This study analyses important lagomorph cranial characters that have been underutilized (i.e. fenestration) or not considered (i.e., ICJ and Facial Tilt) in previous phylogenetic studies. The primary need for this manuscript is the need for more detail regarding the characters discussed, which I note elsewhere with more specific recommendation.

Please see additional annotated comments I have made to the of version of this manuscript.

---

## Round 0.2 · Minor Revisions

I believe that the revised version of the manuscript has improved considerably.
The authors have addressed all reviewers’ suggestions and have modified the manuscript accordingly.

However, Reviewer 1 believes there is still a major issue regarding the definition of character 113. Please take into account the reviewer's suggestion of modifying and simplifying the definition of this character to make it clearer to a wider readership.

Reviewer 1’s additional comments (minor and formatting issues) should also be addressed.

Reviewer 1 ·

Basic reporting

The revised version of the manuscript is an improvement over the previous version, but it also contains problems that should be addressed prior to final acceptance of the article for publication.

Experimental design

Major issue: the definition of character 113

The authors modified the definition of character 113 in response to my previous comments, they have also clearly illustrated the specific character states, making the distinction between them much clearer for the readers. However, the proposed definitions contain a huge error that likely arose from a misunderstanding of one of Dr Kraatz’s (Reviewer 3) comments.

In his comments to the reviewers, Dr Kraatz noted that ‘the lacrimal duct, which has a bony remnant, passes dorsoposteriorly to ventrolaterally along the lateral rostrum’ and that it ‘seems to be a major boundary for fenestration patterns’. This is a very simplified and somewhat imprecise anatomical description – what Dr Kraatz must have meant was that in lagomorphs, the nasolacrimal (not lacrimal) duct* has a dedicated canal** (referred to by Dr Kraatz as a ‘bony remnant’) that passes anteroventrally (alt. posterodorsally) and anteromedially along the medial wall of the maxilla and premaxilla, respectively (see Frahner 1999:fig. 1B and Wolniewicz and Fostowicz-Frelik 2021:figs 5B, 6B) and that the position of this canal on the medial wall of the maxilla reflects the boundary between the ‘reduced’ and ‘advanced’ fenestration character states visible on the maxilla in lateral view. The authors must have not understood this, as the structure they described and highlighted in Fig. 1D–F as the ‘bony remnant of the lacrimal duct’ in reality represents the posterior part of the maxillary corpus (=orbital process of maxilla of Craigie [1948]) (Wolniewicz and Fostowicz-Frelik 2021).

Because Dr Kraatz’s explanation was arguably too simplified and imprecise and was not correctly understood by the authors, and because using a structure visible in medial view to distinguish between character states visible in lateral view is problematic, I suggest the authors modify and simplify their definition of character states for char. 113, so that they are clear for a broad readership, especially those readers not specialised in lagomorph (and more broadly mammal) anatomy. Judging from Fig. 1D–F, one such simple definition could be as follows:

Character 113 lateral fenestration of maxilla (if present)--- (0) large single opening (e.g., Ochotona pallasi) (Fig. 1D), (1) a latticework of small openings restricted to the dorsal (or posterodorsal – whichever the authors feel describes the state better) part of the maxillary corpus (reduced) (e.g., Caprolagus hispidus) (Fig. 1E), (2) an extensive latticework of small openings occupying the dorsal (posterodorsal) as well as the ventral (anteroventral) surface of the maxillary corpus (advanced) (e.g., Lepus timidus) (Fig. 1F).

*in anatomy, ‘duct’ is used to describe the soft-tissue ‘tube’ that acts as passage for secretions/excretions associated with specific organs

** ‘canal’ is the correct anatomical term used for structures that are located within the bone and act as passage for the soft-tissue duct

Validity of the findings

no comment

Additional comments

Minor issues:

1) To maintain consistency and clarity in differentiating between extinct and extant taxa, I recommend the authors use the ‘†’ symbol to indicate extinct taxa not only in Table 1 and the ancestral state reconstruction figures, but also in the main manuscript text, remaining figures and figure captions.

2) In Table 1, please separate the third column into two separate columns: literature used to score characters and specimens used to score characters. Most of the specimen tags in the table do not represent actual specimen numbers, but file/project names on Morphosource and other repositories. Please ask data uploaders or museum curators about correct specimen numbers and update the table accordingly. Scoring specimens based on personal collections (Oryctolagus) should be avoided. I also do not understand how character scores for new characters were ‘pre-existing’ for some taxa.

Formatting issues:

Line 51: Cretaceous–Palaeogene represents a temporal range, so the hyphen should be replaced with an en-dash (see also lines 113, 167, 172).

Line 65: following the majority of in-text references, the author names Petajan, Songster and McNeil should be separated from the date 1981 by a comma (see also lines 67, 125, 289).

Line 75: please reference Fig. 1G of Wood-Bailey’s manuscript after Bramble (1989).

Line 85: separate the author names from the date with a space in the in-text citation.

Lines 91–92: in-text citations should be separated with a semicolon, not a comma.

Line 192: insert space between sentences (see also line 243).

Line 200: the name Nesolagus should be in italics (see also line 260).

Line 249: delete first ‘lateral’.

Line 250: close bracket.

Lines 274–276: The authors state that ‘the estimates for the divergence of leporids are consistent within all three phylogenies’, but they cover a range of 5 million years, which arguably represents quite a long time span. I understand the authors wanted to note that the leproid divergence estimates are more well-constrained in comparison to the leporid/ochotonid split age estimates?

Line 321: delete the spaces separating en-dashes and numbers in both brackets.

Line 341: space between 23 km.

Lines 401–403: Anas should be written with an upper-case A and Anas platyrhynchos should be in italics; please make sure all other references are correctly formatted.

Figure 1: replace hyphens with en-dashes for panel ranges in caption (e.g. A–C instead of A-C); modify ‘bony remnant of lacrimal duct’ (see above).

Figure 2: consider adding the ‘†’ symbol to denote extinct taxa.

Figure 4: modify ‘bony remnant of lacrimal duct’ in caption (see above).

Table 1: fix inappropriate bracket in Meng et al. citation in caption; ‘outgroup for Glires’ and ‘scored characters pre-existing’ should not be in italics.

Table 2: replace hyphens with en-dashes and delete spaces between en-dashes and numbers in ‘divergence calibration’ column.

·

Basic reporting

ok

Experimental design

ok

Validity of the findings

ok

---

## Round 0.3 · accepted · Accept

Thank you for addressing the reviewers' comments. Your attention to these last details is appreciated. I now believe that your manuscript is suitable for publication. Congratulations!